# Mesoscopic fluctuations in entanglement dynamics

Lih-King Lim [1], Cunzhong Lou[1] & Chushun Tian [2] ✉

Understanding fluctuation phenomena plays a dominant role in the development of many-body physics. The time evolution of entanglement is essential to a broad range of subjects in many-body physics, ranging from exotic quantum matter to quantum thermalization. Stemming from various dynamical processes of information, fluctuations in entanglement evolution differ conceptually from out-of-equilibrium fluctuations of traditional physical quantities. Their studies remain elusive. Here we uncover an emergent random structure in the evolution of the many-body wavefunction in two classes of integrable—either interacting or noninteracting—lattice models. It gives rise to out-of-equilibrium entanglement fluctuations which fall into the paradigm of mesoscopic fluctuations of wave interference origin. Specifically, the entanglement entropy variance obeys a universal scaling law in each class, and the full distribution displays a sub-Gaussian upper and a sub-Gamma lower tail. These statistics are independent of both the system's microscopic details and the choice of entanglement probes, and broaden the class of mesoscopic universalities. They have practical implications for controlling entanglement in mesoscopic devices.

When an isolated many-body system evolves, entanglement tends to spread. Owing to the diversity of the fate of the wavefunction evolution (e.g., localized or delocalized, thermalized or not thermalized), a wealth of entanglement patterns develop[1–7]. These patterns are the building blocks of the physics of recently discovered exotic phases of matter[4,7,8], and are central to the foundations of statistical mechanics[6,7]. Understanding the long-time evolution of entanglement, and especially its universal aspects, is indispensable in the study of pattern formation.

To address this issue, one often investigates mesoscopic rather than macroscopic systems. Recent advancements in quantum simulation platforms, ranging from cold atoms, and trapped ions to superconducting qubits, have made possible the measurement of information-theoretic observables and the experimental study of entanglement evolution[6,7,9]. In these investigations, quantum coherence is maintained across the entire sample, as required also for mesoscopic electronic and photonic devices[10,11]. At the same time, the relationship between the evolution of entanglement and quantum

thermalization in isolated systems is currently under investigations[6,7]. Since various scenarios for the latter[12–17] are built upon a basis of wavefunctions with finite spatial extent, emphasis has naturally been placed on the dynamics of entanglement on the mesoscopic scale.

A prominent feature of mesoscopic systems is the occurrence of unique fluctuation phenomena when randomness due to quenched disorders[10,11] or chaos[18,19] is present. Notably, the conductance—a basic probe of mesoscopic transport—fluctuations have a universal variance, independent of sample size and the strength of randomness[20,21]. Mesoscopic fluctuations are of wave interference origin and are conceptually different from thermodynamic fluctuations. They are related to various entanglement properties[22,23]. The universality of these fluctuations is at the heart of mesoscopic physics.

In fact, there is a rapid increase in interest in entanglement fluctuations. In particular, understanding out-of-equilibrium entanglement fluctuation properties is key to the statistical physics of isolated systems[24,25]. So far studies have focused on the kinematic case[16,26–29], where fluctuations arise from random sampling of some pure state

¹School of Physics, Zhejiang University, 310027 Hangzhou, Zhejiang, China. ²CAS Key Laboratory of Theoretical Physics and Institute of Theoretical Physics, Chinese Academy of Sciences, 100190 Beijing, China. ✉e-mail: ct@mail.itp.ac.cn

ensemble, initiated by Page[26]. On one hand, since kinematic theories cannot describe wave effects and dynamical properties of the Schrödinger evolution[30], out-of-equilibrium entanglement fluctuations are beyond the framework of those theories. On the other hand, there have been big efforts on out-of-equilibrium fluctuations in isolated quantum systems[31–38]. But so far focus has been on traditional physical quantities, and little has been known about information-theoretic observables such as the entanglement entropy and the Rényi entropy[25,39].

Here, we develop an analytical theory for long-time dynamics of entanglement in two classes of integrable lattice models. One class of models, including the Rice–Mele model and the transverse field Ising chain, can be mapped to noninteracting fermions; the other class includes interacting spin chains, with the spin-1/2 Heisenberg *XXZ* model as a representative. Our theory relies crucially on the uncovering of a random structure emergent from the dynamical phases in the wavefunction evolution. Treating various information-theoretic observables as unconventional mesoscopic probes, we show that their out-of-equilibrium fluctuations fall into the paradigm of universal mesoscopic fluctuations in disordered or chaotic systems. Our findings have immediate implications for controlling entanglement in quantum simulation platforms.

## Results

### Description of main results

**Emergent mesoscopic fluctuations.** We find that the many-body wavefunction evolution endows the correlation matrix (the reduced density matrix) with a random structure for noninteracting (interacting) models, even though the system is neither chaotic nor disordered. Specifically, for noninteracting models the time dependence enters through $N \approx \frac{L}{2}$ dynamical phases $(\omega_1 t, ..., \omega_N t) \equiv \boldsymbol{\omega} t$, with $L$ being the

number of unit cells, so that the instantaneous correlation matrix $C(t)$ is given by some $N$-variable (matrix-valued) function $\tilde{C}(\boldsymbol{\varphi})$ for $\boldsymbol{\varphi} = \boldsymbol{\omega} t$; due to the incommensurability of $\boldsymbol{\omega}$ an ensemble of random matrices $\tilde{C}(\boldsymbol{\varphi})$ then results. Each $\tilde{C}(\boldsymbol{\varphi})$ is determined by $\boldsymbol{\varphi}$, the virtual disorder realization uniformly distributed over a $N$-dimensional torus (Fig. 1). It describes a virtual disordered sample, and determines entanglement properties of that sample in the same fashion as $C(t)$ determines the system's instantaneous entanglement properties. For interacting models, $C(t)$ and $\tilde{C}(\boldsymbol{\varphi})$ are replaced by the instantaneous reduced density matrix $\rho_A(t)$ and its $N$-variable counterpart $\tilde{\rho}_A(\boldsymbol{\varphi})$, respectively, and $N$ grows exponentially with $L$. So, when the system's wavefunction evolves, the trajectory $\boldsymbol{\varphi} = \boldsymbol{\omega} t$ sweeps out the entire disorder ensemble, trading the temporal fluctuations of various information-theoretic observables to mesoscopic *sample-to-sample* fluctuations[20,21]. In particular, we find that these out-of-equilibrium entanglement fluctuations arise from wave interference, similar to mesoscopic fluctuations. Interestingly, this kind of trajectory plays an important role in Chirikov's studies of the relations between mesoscopic physics and quantum chaos[40].

However, there are important differences between ordinary quenched disorders and the randomness emergent from entanglement evolution. As shown below, the latter has a strength $\sim 1/\sqrt{L}$ for noninteracting models and $\sim e^{-L}$ for interacting, and thus diminishes for $L \to \infty$. This situation renders canonical mesoscopic theories based on diagrammatical[10,11] and field-theoretical[41] methods inapplicable, since they require the disorder strength to be independent of the sample size. In addition, because $C(t)$ is a (block-)Toeplitz matrix and very little[42] is known about the spectral statistics of random Toeplitz matrices, mesoscopic theories based on random matrix methods[43] are inapplicable either. Here we develop a different approach based on the modern nonasymptotic probability theory[44], that relies merely on the

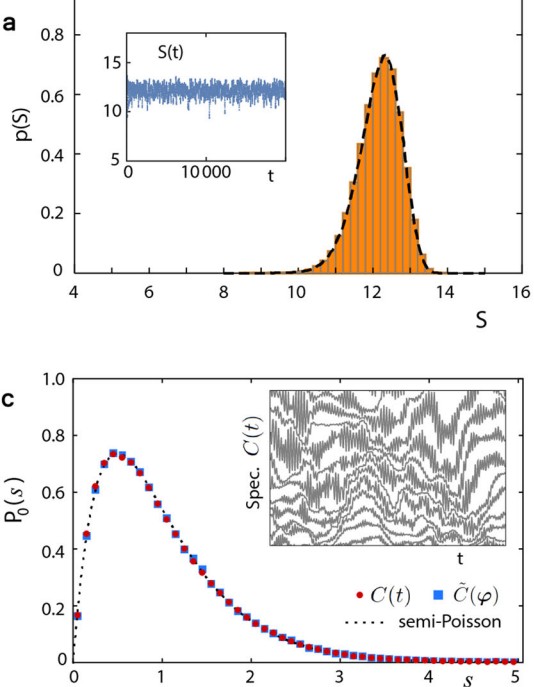

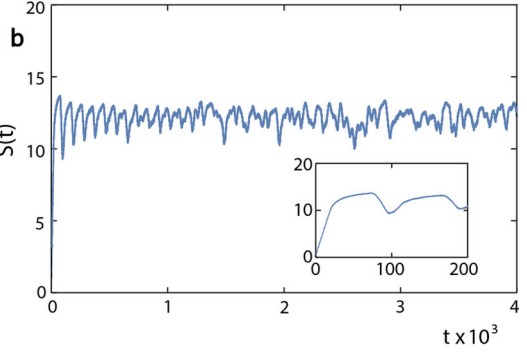

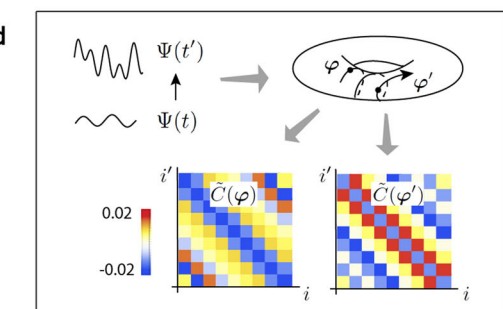

**Fig. 1 | Emergence of mesoscopic fluctuations in entanglement evolution. a** We simulate entanglement entropy evolution $S(t)$ of Rice–Mele model up to $t = 10^4$ in unit of $\hbar/J$ ($L_A = 25$, $L = 124$); see the inset. Its fluctuation statistics (histograms) is shown to be equivalent to the statistics of entanglement entropy fluctuations in an ensemble of virtual disordered samples (dashed line, for $5 \times 10^5$ disorder realizations $\boldsymbol{\varphi}$). **b** These long-time fluctuations differ from the profile of $S(t)$ at early time. Inset: At early time $S(t)$ exhibits growth followed by damped oscillations.

**c** Simulations show that the nearest-neighbor spacing distribution characterizing spectral fluctuations of the correlation matrix $C(t)$ (inset) is indistinguishable from that for an ensemble of truly random matrices $\tilde{C}(\boldsymbol{\varphi})$, and is semi-Poissonian.
**d** Physically, as system's wavefunction evolves, the dynamical phases $\boldsymbol{\varphi} = (\omega_1 t, ..., \omega_N t)$ sweeps out an ensemble of mesoscopic samples $\tilde{C}(\boldsymbol{\varphi})$. The quench protocol is $(J, J', M) : (1, 0.5, 0.5) \to (1, 1.5, 1.5)$.

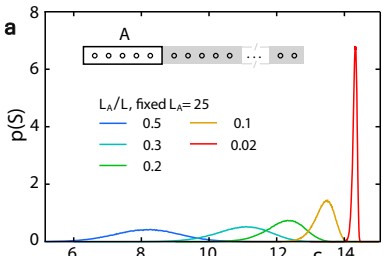
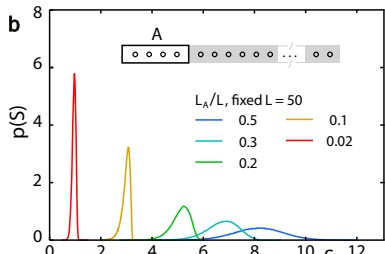
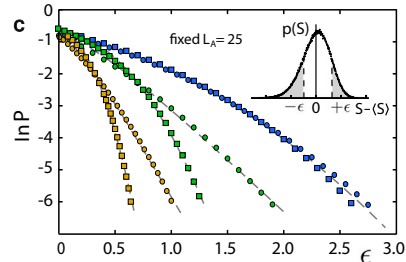

**Fig. 2 | Entanglement entropy distribution of Rice-Mele model.** We perform statistical analysis of the temporal fluctuations in the simulated entanglement entropy evolution. **a** Variation of the distribution with increasing $L$ at fixed $L_A$. **b** Same as a., but with increasing $L_A$ at fixed $L$. **c** The large deviation probability $\mathbf{P}(|S$ $-\langle S\rangle| \geq \epsilon)$, with upper and lower tail respectively, is well fitted by Eq. (2) (dashed lines), implying that the upper (squares) tail distribution is sub-Gaussian and the lower (circles) is sub-Gamma. The ratio $L_A/L$ is 0.1 (yellow), 0.2 (green) and 0.5 (blue), with the same quench protocol as Fig. 1.

statistical independence of the components of $\boldsymbol{\varphi}$ and applies to any $L$. A related approach has recently been used to find novel universalities in mesoscopic transport[45].

**Fluctuation statistics.** Uncovering the random structure, we show that fluctuations in entanglement evolution exhibit intriguing universal behaviors for each class of models, independent of microscopic details. First, when the variance $\mathrm{Var}(S)$ of the entanglement entropy $S$, as well as $L$ and $L_A$ (the subsystem size), are rescaled by appropriate microscopic quantities, the universal scaling law:

$$\mathrm{Var}(S) = \begin{cases} 1/L + L_A^3/L^2, & \text{for noninteracting} \\ L_A^\beta e^{-L}, & \text{for interacting} \end{cases} \quad (1)$$

models follow, where $\beta$ is between 2 and 4, depending on both the initial state and the system's parameters. Second, the distribution of $S$ has a universal shape, and is asymmetric with respect to its mean $\langle S\rangle$, displaying a sub-Gaussian upper and a sub-Gamma lower tail. In particular, for both classes, the probability of large deviation $\epsilon$ is

$$\mathbf{P}(|S - \langle S\rangle| \geq \epsilon) = \begin{cases} e^{-\frac{\epsilon^2}{2b_+}}, & \text{for } S - \langle S\rangle > 0 \\ e^{-\frac{\epsilon^2}{2(b_- + c\epsilon)}}, & \text{for } S - \langle S\rangle < 0 \end{cases}, \quad (2)$$

where $\mathfrak{b}_\pm \propto \mathrm{Var}(S)$ and $\mathfrak{c} > 0$ depend on the ratio $L_A/L$. Third, Eqs. (1) and (2) hold for other probes, e.g., the Rényi entropy. These universal fluctuation behaviors are irrespective of the location of $\langle S\rangle$ in Page's curve[26]. By Eq. (1) at fixed $L_A$ the variance vanishes in the limit $L \to \infty$ (cf. Fig. 2a), implying the full suppression of temporal fluctuations beyond some critical time, in agreement with a benchmark result of entanglement evolution[1] and for the first time generalizing that result to interacting spin models analytically. How the entanglement entropy saturates in interacting systems is crucial to understanding experiments on entanglement dynamics[6,7]. In contrast, at fixed $L$, as $L_A$ increases the variance displays a power-law growth (cf. Figs. 2b and 5c), which is faster than $\sim L_A$ displayed by typical extensive quantities. We shall see below this enhanced growth results from quantum interference.

## Theory and numerical verifications

**Noninteracting models.** Having summarized the main results, we outline the derivations and present numerical verifications. A complete description is given in Supplementary Notes 1–8 for non-interacting models and Supplementary Note 9 for interacting models. We start from the free fermion case, and focus on the Rice–Mele model

with the Hamiltonian (see Supplementary Note 1)

$$H_{\mathrm{RM}} = -\sum_{i=1}^{L} \left( J c_{i\bar{A}}^\dagger c_{i\bar{B}} + J' c_{i\bar{B}}^\dagger c_{(i+1)\bar{A}} + \text{h.c.} \right)$$
$$+ M \sum_{i=1}^{L} \left( c_{i\bar{A}}^\dagger c_{i\bar{A}} - c_{i\bar{B}}^\dagger c_{i\bar{B}} \right). \quad (3)$$

Here $J, J'$ are the hopping amplitudes, $M$ is the staggered onsite mass, $c_{i\sigma}^\dagger, c_{i\sigma}$ ($\sigma = \bar{A}, \bar{B}$) are, respectively, fermionic creation and annihilation operators at the $\sigma$-sublattice sites belonging to the $i$th unit cell. The system has a total of $L$ unit cells and is subjected to the periodic boundary condition. Generalizations to other models mappable to free fermions are straightforward. Let the system be at the half-filling ground state $\Psi(0)$. At $t = 0$, we suddenly change the parameters of the Hamiltonian. So the pre-quench state $\Psi(0)$ evolves unitarily under the new Hamiltonian to state $\Psi(t)$ at a later time $t$. Because $\Psi(0)$ is a Gaussian state and the system is fermionic, the instantaneous entanglement entropy can be expressed as

$$S(t) = \int d\lambda\, e(\lambda)\, \mathrm{Tr}_A\, \delta(\lambda - C(t)) \quad (4)$$

using the method in refs. 46–49. Here $e(\lambda) = -\lambda \ln \lambda - (1-\lambda)\ln(1-\lambda)$ is the binary entropy function. $\mathrm{Tr}_A\, \delta(...)$ gives the spectral density of the correlation matrix $C(t)$ with element $C_{i\sigma, i'\sigma'}(t) = \langle \Psi(t)|c_{i\sigma}^\dagger c_{i'\sigma'}|\Psi(t)\rangle$. The trace is restricted to the subsystem A. When replacing $e(\lambda)$ by an appropriate function of $\lambda$, we obtain other entanglement probes such as the Rényi entropy. This kind of expression indicates that the evolving spectral density underlies out-of-equilibrium behaviors of different entanglement probes. They are analogous to the expressions for probes of mesoscopic transport. Indeed, if we replace $C(t)$ with the product of the transmission matrix and its hermitian conjugate, we transform Eq. (4) to the Landauer formula for conductance with $e(\lambda)$ changed to $\lambda$, and to formulas for other transport probes with $e(\lambda)$ changed to appropriate functions of $\lambda$[43].

Because the eigenenergy spectrum displays a reflection and a particle-hole symmetry, when particle eigenenergies $\frac{\omega_m}{2}$ (Planck's constant set to unity) at Bloch momenta $k_m = \frac{2\pi(m-1)}{L}, m = 1,...,N = [\frac{L}{2}] + 1$, are given, all other particles and all hole eigenenergies are known. Due to the translational invariance of the system, the time parameter enters the correlation matrix through the dynamical phases $\boldsymbol{\omega}t$ associated with $\boldsymbol{\omega} \equiv (\omega_1, ..., \omega_N)$. Specifically, we can define a function $\tilde{C}(\boldsymbol{\varphi}) = C_0 + C_1(\boldsymbol{\varphi})$ on the $N$-dimensional torus. Leaving its detailed form for Supplementary Note 2, here we only expose its key properties. First, $C_{0,1}$ are block-Toeplitz matrices, with elements $(C_{0,1})_{ii'}$ being $2 \times 2$ blocks defined in the sublattice sector and depending on the unit cell indexes $i, i'$ via $(i - i')$, i.e., $(C_{0,1})_{ii'} \equiv (C_{0,1})_{i-i'}$. Second, $C_0$ is $\boldsymbol{\varphi}$-independent, whereas $C_1$ is not and its elements take

the form of

$$(C_1)_l \equiv \frac{1}{L} \sum_{m=1}^{N} \left( R_l(k_m) \cos \varphi_m + I_l(k_m) \sin \varphi_m \right), \qquad (5)$$

where the elements of blocks, $R$'s, $I$'s, are complex and depend on $k_m$ (as well as post-quench Hamiltonian parameters). Then $C(t)$ is given by $\tilde{C}(\boldsymbol{\varphi})$ at $\boldsymbol{\varphi} = \boldsymbol{\omega} t$. Similarly, with the introduction of $S(\boldsymbol{\varphi}) \equiv \int d\lambda \, e(\lambda) \, \mathrm{Tr}_A \, \delta(\lambda - \tilde{C}(\boldsymbol{\varphi}))$ in parallel to Eq. (4) (for notational simplicity we use the same symbol $S$ despite differences in the arguments.), $S(t)$ is given by $S(\boldsymbol{\varphi})$ at $\boldsymbol{\varphi} = \boldsymbol{\omega} t$. This implies that, like $C(t)$, an evolving entanglement probe depends on $t$ through the dynamical phases $\boldsymbol{\omega} t$. Such dependence has an immediate consequence (see Supplementary Note 3 for details). That is, because in general, the components of $\boldsymbol{\omega}$ are incommensurate, after initial growth[1] and damped oscillations[50] due to the traversal of quasiparticle pairs or the incomplete revival of wavefunction (Fig. 1b), an entanglement probe displays quasiperiodic oscillations (Fig. 1a, inset), which are reproducible under the same initial conditions.

To understand the fluctuation properties of quasiperiodic oscillations we note that the trajectory $\boldsymbol{\varphi} = \boldsymbol{\omega} t$ generates an ensemble of random matrices $\tilde{C}(\boldsymbol{\varphi})$, each of which is determined by the disorder realization, $\boldsymbol{\varphi}$, and thus is separated into two parts: nonrandom $C_0$ and random $C_1(\boldsymbol{\varphi})$. The probability measure of this ensemble is induced by the uniform distribution of $\boldsymbol{\varphi}$ via Eq. (5). This ensemble has some prominent features (see Supplementary Note 4 for detailed discussions): First, since $\varphi_m$'s are statistically independent, Eq. (5) implies that each element randomly fluctuates around its mean, with a magnitude $\sim 1/\sqrt{L}$. Thus for fixed ratio $L_A/L$, the randomness diminishes in the limit of large matrix size. Second, the elements of two distinct blocks are statistically independent. Third, the average elements decay rapidly with their distance to the main diagonal. These features lead to a semi-Poissonian nearest-neighbor spacing distribution[51,52],

$$P_0(s) = 4s e^{-2s}, \qquad (6)$$

as shown in simulations (Fig. 1c); this kind of universal intermediate statistics was originally found for Anderson transitions[53]. Strikingly, despite that the Rice–Mele model is integrable and has no extrinsic randomness, the evolving correlation matrix can exhibit level repulsion: $P_0(s \to 0) \sim s$, which is a distinctive property of quantum chaos[18,19]. We can demonstrate that the statistical equivalence of the ensemble of $\tilde{C}(\boldsymbol{\varphi})$ and the time series $C(t)$ (Fig. 1c) hinges only on the incommensurabilty of $\boldsymbol{\omega}$ (see Supplementary Note 8 when this condition is not met). Furthermore, much like that a transmission matrix determines transport properties of a mesoscopic sample, a matrix $\tilde{C}(\boldsymbol{\varphi})$ determines $S(\boldsymbol{\varphi})$ and other entanglement probes of a virtual mesoscopic sample at the disorder realization $\boldsymbol{\varphi}$; consequently, the statistical equivalence between $C(t)$ and $\tilde{C}(\boldsymbol{\varphi})$ leads to the statistical equivalence between out-of-equilibrium and sample-to-sample fluctuations of various entanglement probes, in agreement with simulation results (Fig. 1a).

Exploiting this equivalence, we proceed to study the statistics of entanglement entropy fluctuations (see Supplementary Notes 5 and 6 for full details). To overcome the difficulties discussed in the introduction with the unusual disorder structure, below we combine the continuity properties of the $N$-variable function $S(\boldsymbol{\varphi})$ with the non-asymptotic probabilistic method, so-called concentration inequality[44]. This allows us to work out a statistical theory for mesoscopic sample-to-sample fluctuations of $S(\boldsymbol{\varphi})$ at total system size $L$, which is *finite* so that the disorder strength does not vanish.

In order to study the distribution of $S(\boldsymbol{\varphi})$, we introduce the logarithmic moment-generating function $G(u) \equiv \ln\langle e^{u(S-\langle S \rangle)} \rangle$, with $u$ being real and $\langle \cdot \rangle$ denoting the average over $\boldsymbol{\varphi}$. Consider the downward fluctuations (i.e., $S - \langle S \rangle < 0$) first. Because the $N$ components of $\boldsymbol{\varphi}$ are

statistically independent, we can apply the so-called modified logarithmic Sobolev inequality[44] to obtain

$$\frac{d}{du} \frac{G}{u} \leq \frac{1}{u^2} \frac{\left\langle \left[ \sum_{m=1}^{N} e^{u(S-\langle S \rangle)} \phi(-u(S-S_m^-)) \right] \right\rangle}{\left\langle e^{u(S-\langle S \rangle)} \right\rangle} \qquad (7)$$

with $\phi(x) = e^x - x - 1$ and $u \leq 0$. Here $S_m^-$ is the maximal value of $S(\boldsymbol{\varphi})$, when $\varphi_m$ varies and other arguments are fixed. Observing that the leading $u$-expansion of the right-hand side is $\frac{b_-}{2}$, with

$$b_- \equiv \sum_{m=1}^{N} \left\langle (S-S_m^-)^2 \right\rangle, \qquad (8)$$

we separate the right-hand side of the inequality into two terms, $\frac{b_-}{2}$ and the remainder. Then, we show that the latter is bounded by $c_- \frac{dG}{du}$ with $c_-$ being a negative constant. So we cast inequality (7) to

$$\frac{d}{du} \frac{(1+|c_-|u)G}{u} \leq \frac{b_-}{2}, \qquad (9)$$

which can be readily integrated to give $G \leq \frac{b_-}{2} \frac{u^2}{1+|c_-|u}$. Such bound holds also for Gamma random variables. It generalizes the tail behaviors of the Gamma distribution, giving the so-called sub-Gamma tail[44]. Specifically, following standard procedures, we can use Markov's inequality to turn this bound for $G$ into a bound for the probability of downward fluctuations. The result is

$$\mathbf{P}(S < \langle S \rangle - \epsilon) \leq e^{-\epsilon^2 / 2(b_- + |c_-|\epsilon)} \qquad (10)$$

for any $\epsilon > 0$. This gives a sub-Gamma lower tail, which crosses over from a Gaussian to an exponential form at $\epsilon \sim b_-/|c_-|$.

Similarly, we can study the upward fluctuations (i.e., $S - \langle S \rangle > 0$). We replace $S_m^-$ in inequality (7) with $S_m^+$, which is the minimal $S(\boldsymbol{\varphi})$ when $\varphi_m$ varies and other arguments are fixed, and consider $u > 0$. Upon separating $\frac{b_+}{2}$, with $b_+ \equiv \sum_{m=1}^{N} \langle (S-S_m^+)^2 \rangle$, from the right-hand side of the inequality, the remainder is negative. As a result, $c_-$ is replaced by 0 and $G \leq \frac{b_+ u^2}{2}$, giving

$$\mathbf{P}(S > \langle S \rangle + \epsilon) \leq e^{-\epsilon^2 / (2b_+)} \qquad (11)$$

for any $\epsilon > 0$, which is a sub-Gaussian upper tail.

The inequalities (10) and (11) show that $S(\boldsymbol{\varphi})$ concentrates around $\langle S \rangle$ albeit with different bounds for upward and downward fluctuations. Simulations further show that the exact deviation probability for large downward (upward) fluctuations agrees with the form given by the right-hand side of the corresponding concentration inequality, with $b_\pm$ and $c_-$ as fitting parameters (Fig. 2c). Therefore, for large deviation, the upper (lower) tail distribution has the universal form given by the first (second) line in Eq. (2), and the parameters $\mathfrak{b}_\pm$ and $\mathfrak{c}$ in Eq. (2) are proportional to $b_\pm$ and $c_-$, respectively. So for large $\epsilon$ the upper tail is always Gaussian $e^{-\epsilon^2/(2b_+)}$ while the lower is always exponential $e^{-\epsilon/(2c)}$, different from the distribution tails of thermodynamic fluctuations which are symmetric and Gaussian.

With Eq. (2) we find that the variance $\mathrm{Var}(S)$ is given by $b_\pm$. To calculate the latter, note that by the mean value theorem, there exists $\bar{\varphi}_m$ between $\varphi_m$ and $\varphi_m^\pm$ (at which $S_m^\pm$ is reached), so that $(S-S_m^\pm)^2$ is given by $(\varphi_m - \varphi_m^\pm)^2 (\partial_{\bar{\varphi}_m} S)^2$. Then, for large $L$ the Fourier series of $\partial_{\varphi_m} S$ with respect to $\varphi_m$ is truncated at the second harmonics, giving $(\partial_{\bar{\varphi}_m} S)^2 \sim \int \frac{d\varphi_m}{2\pi} (\partial_{\varphi_m} S)^2$. Applying these analyses to the definitions of $b_\pm$, we obtain

$$\mathrm{Var}(S) \propto \left\langle |\partial_{\boldsymbol{\varphi}} S|^2 \right\rangle. \qquad (12)$$

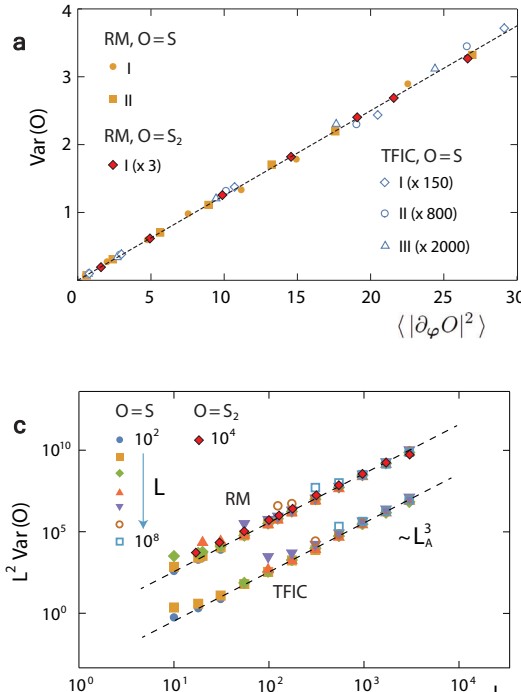

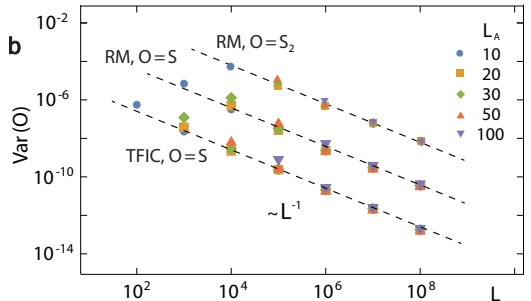

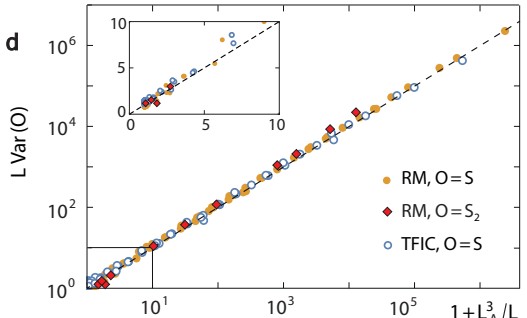

**Fig. 3 | Universal scaling behaviors of the variance in noninteracting models.**
We perform simulations for both Rice-Mele (RM) model and transverse field Ising chain (TFIC), whose Hamiltonian $H_{TFIC} = (-1/2)\sum_{i=1}^{L}(\sigma_i^x \sigma_{i+1}^x + h\sigma_i^z)$ with $\sigma_i^\alpha$ the Pauli matrices and $h$ the external magnetic field, for different sizes and different quench protocols to study the variance of two entanglement probes $O = S, S_2$. **a** For both $O$ the data confirm the relation (12). **b-c** They also confirm the limiting scaling behavior described by the first (second) term of the first line of Eq. (1) for

sufficiently small (large) $L_A^3/L$. **d** For different models, after rescaling Var($O$), $L$, $L_A$ all data collapse to the universal curve described by Eq. (1) for noninteracting models. Inset: Zoom-in view of the regime near the origin. All theoretical predictions are presented by dashed lines. The quench protocols of $(J, J', M)$ for RM are $I$: $(1, 0.7, 0.3) \to (0.3, 1.0, 0.001)$ and $II$: $(0.6, 1.0, 0.8) \to (1.0, 0.2, 10)$, and for TFIC are $I$: $h = 3 \to 2$; $II$: $h = 3 \to 5$ and $III$: $h = 5 \to 2$.

This relation is confirmed numerically (Fig. 3a), and the proportionality coefficient is found to be $\approx 1/8$. Equation (12) uncovers a relation between entanglement entropy fluctuations and continuity properties of the $N$-variable function $S(\boldsymbol{\varphi})$. It resembles the so-called concentration-of-measure phenomenon, a modern perspective of probability theory[54,55], where fluctuations of an observable are controlled by its *Lipschitz continuity*. This continuity is a key ingredient of universal wave-to-wave fluctuations in mesoscopic transport[45].

By definition of $S(\boldsymbol{\varphi})$, we have $\partial_{\varphi_m} S = \mathrm{Tr}_A(\ln(\tilde{C}^{-1} - \mathbb{I})\partial_{\varphi_m} C_1)$. Because of $C_1 = \mathcal{O}(1/\sqrt{L})$, we expand the logarithm in $C_1$ up to the first order. Taking into account that $C_0$ is short-ranged, we obtain

$$\partial_{\varphi_m} S = -\mathrm{Tr}_A\left[(H_0 + (\partial_{C_0} H_0)C_1)\partial_{\varphi_m} C_1\right], \tag{13}$$

where $H_0 = \ln(C_0^{-1} - \mathbb{I})$ is the entanglement Hamiltonian in the absence of disorder. Substituting Eq. (13) into Eq. (12), we find that the two terms in Eq. (13) contribute to the variance separately. The contribution by the first term is $a/L$ and that by the second is $bL_A^3/L^2$, and the former (latter) is found to be a subsystem's edge (bulk) effect. Here the coefficient $a$ is proportional to the square of the size of the subsystem's edge, and both $a$ and $b$ have no dependence on $L, L_A$. Upon rescaling: $L, L_A$ by $\sqrt{a/b}$ and Var($S$) by $\sqrt{ab}$, we obtain the scaling law (1) for noninteracting models, which is confirmed by simulations (Fig. 3b-d). By Eq. (1), one enters the regime Var($S$) $= L^{-1}$ for $L_A \ll L^{1/3}$ (b) and the regime Var($S$) $= L_A^3/L^2$ for $L_A \gg L^{1/3}$ (c).

Let us consider other entanglement probes such as the second-order Rényi entropy $S_2$. As said above, in this case, we have an expression similar to Eq. (4), with $e(\lambda)$ changed (see Supplementary Note 2). Repeating the analysis above, we find for $S_2$ the same relation as (12). Furthermore, we can calculate $\langle|\partial_{\boldsymbol{\varphi}} S_2|^2\rangle$ in the same way as $\langle|\partial_{\boldsymbol{\varphi}} S|^2\rangle$. As a result, we find that Var($S_2$) obeys the

same scaling law as Eq. (1) for noninteracting models. These statistics of $S_2$ are confirmed numerically (Fig. 3). In Supplementary Notes 5–7 we further show that Eqs. (1), (2) and (12) hold for more general probes.

To understand physically the scaling behavior we use the concept of coherent entangled quasiparticle pair[1]. Consider a quasiparticle inside the subsystem A. When pairing with another outside, it contributes to the bipartite entanglement. Due to the Heisenberg uncertainty, this particle's position fluctuates with time, leading to the temporal fluctuation $\Phi(t)$ of the pairing amplitude. In the simplest case, the particle hops virtually from a site $i$ to $j$ (in A as well) and back to $i$. Since the entangled pair is a correlation effect, $\Phi(t) \sim \sum_{ij}(C_1(\boldsymbol{\omega}t))_{ij}(C_1(\boldsymbol{\omega}t))_{ji}$ and thus by Eq. (5) $\Phi(t) \sim \frac{1}{L^2}\sum_{ij}\sum_{mn}e^{i(k_m-k_n)(i-j)}e^{i(\omega_m+\omega_n)t/2}$, where $k_m, \frac{\omega_m}{2}$ are respectively the Bloch momentum and the particle eigenenergy associated with the hopping $i \to j$, and $k_n, \frac{\omega_n}{2}$ with $j \to i$. The variance of a generic entanglement probe is given by

$$\int dt|\Phi(t)|^2 \sim \frac{1}{L^4}\sum_{iji'j'}\sum_{mnm'n'}\delta_{\omega_m+\omega_n,\,\omega_{m'}+\omega_{n'}} \\ \times e^{i((k_m-k_n)(i-j)-(k_{m'}-k_{n'})(i'-j'))}, \tag{14}$$

where $(k_m-k_n)(i-j)$ and $(k_{m'}-k_{n'})(i'-j')$ are the phases of the paths: $i \to j \to i$ and $i' \to j' \to i'$, respectively. Because $\omega$'s are incommensurate, we obtain $(m,n) = (m',n')$ or $(n',m')$. So the first sum is dominated by those terms with two phases being identical. As a result, $\int dt|\Phi(t)|^2 \sim L_A^3/L^2$, with the numerator (denominator) given by the first (second) sum: This is the second term in the first line of Eq. (1). We see that it arises from the constructive interference between the two hopping paths (Fig. 4).

**Interacting models**. For interacting models, the correlation function method and Eq. (4) do not hold in general. What happens then? In this case, we have to retreat back to the more general expression of various information-theoretic observables in terms of the instantaneous reduced density matrix $\rho_A(t)$. Below we generalize the theory above to the spin-1/2 anisotropic Heisenberg XXZ model[56] defined by the Hamiltonian

$$H_{XXZ} = J \sum_{i=1}^{L} (S_i^x S_{i+1}^x + S_i^y S_{i+1}^y + \Delta S_i^z S_{i+1}^z), \quad (15)$$

where $S_i^\alpha$ are spin-1/2 operators, $\Delta$ is the anisotropy parameter, and the periodic boundary condition is imposed; generalizations to other interacting models are possible, which we will not discuss further. For simplicity here we consider the initial state $\Psi(0)$ to be an unpolarized random state; see Supplementary Note 9 for their detailed description. Other $\Psi(0)$ will be studied in that note.

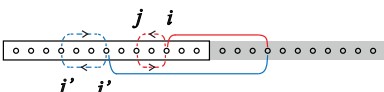

**Fig. 4 | Wave origin of entanglement fluctuations.** The pairing amplitude of the coherent entangled quasiparticle pair (solid lines) fluctuates with time. Constructive interference between two paths due to virtual hopping (blue and red dashed lines), that underlies such fluctuations, leads the variance of a generic entanglement probe to exhibiting the scaling behavior $\sim L_A^3/L^2$ for noninteracting models.

Let $\Psi(0) = \sum_m \chi_m \Psi_m$, where $\Psi_m$'s are eigenstates and $\chi_m$'s are superposition coefficients. The number of excited eigenbases, $D$, may be estimated as the participation ratio $(\sum_m |\chi_m|^4)^{-1}$, which grows exponentially in $L$[36]. As the wavefunction evolves, $\rho_A(t) = \rho_{A0} + \rho_{A1}(t)$, where $\rho_{A0} = \sum_m |\chi_m|^2 \text{Tr}_B |\Psi_m\rangle\langle\Psi_m|$ and $\rho_{A1}(t) = \sum_{m\neq n} e^{-i(\omega_m - \omega_n)t} \chi_m \chi_n^* \text{Tr}_B |\Psi_m\rangle\langle\Psi_n|$ with $\omega_m$'s being eigenenergies, and B is the complement of A. Importantly, all eigenenergy mismatches: $\omega_m - \omega_n$ here are completely determined by $N$ ($=D-1$) mismatches: $\omega_m - \omega_1 \equiv \omega_{m1}$ ($m = 2,...,D$), because of $\omega_m - \omega_n = \omega_{m1} - \omega_{n1}$. So, similar to the free fermion case, the time parameter enters through the $N$ phases $\boldsymbol{\omega}t$, with $\boldsymbol{\omega} \equiv (\omega_{21},...,\omega_{D1})$. Introducing a function $\tilde{\rho}_A(\boldsymbol{\varphi}) = \rho_{A0} + \tilde{\rho}_{A1}(\boldsymbol{\varphi})$ on the $N$-dimensional torus,

$$\tilde{\rho}_{A1}(\boldsymbol{\varphi}) = \sum_{m\neq n} e^{-i(\varphi_m - \varphi_n)} \chi_m \chi_n^* \text{Tr}_B |\Psi_m\rangle\langle\Psi_n|, \quad (16)$$

and associating with $\tilde{\rho}_A(\boldsymbol{\varphi})$ various entanglement probes, e.g. $S(\boldsymbol{\varphi}) = -\text{Tr}_A(\tilde{\rho}_A \ln \tilde{\rho}_A)$ and $S_2(\boldsymbol{\varphi}) = -\ln \text{Tr}_A(\tilde{\rho}_A^2)$, we then obtain corresponding instantaneous probes from $S(\boldsymbol{\varphi})$, $S_2(\boldsymbol{\varphi})$, etc., by setting $\boldsymbol{\varphi} = \boldsymbol{\omega}t$.

The components of $\boldsymbol{\omega}$ are incommensurate in general. Thus various probes: $S(t)$, $S_2(t)$, etc. display quasiperiodic oscillations, which are seen in long-time simulations (by using the standard numerical method[57]). Most importantly, because the trajectory $\boldsymbol{\varphi} = \boldsymbol{\omega}t$ is ergodic on torus, the temporal fluctuations of an entanglement probe say $S(t)$ are statistically equivalent to the sample-to-sample fluctuations of $S(\boldsymbol{\varphi})$, with each sample having a disorder realization $\boldsymbol{\varphi}$ and represented by $\tilde{\rho}_A(\boldsymbol{\varphi})$. Moreover, the sum in Eq. (16) is dominated by $\sim L_A^\mu D^{1+\nu}$ terms, where $1 \leq \mu \leq 2$, $0 \leq \nu < 1$, and the value of $\mu, \nu$ depends on $\Psi(0)$ and $\Delta$ (see Supplementary Note 9 for details). Taking $\chi_m \chi_n^* \sim 1/D$ into

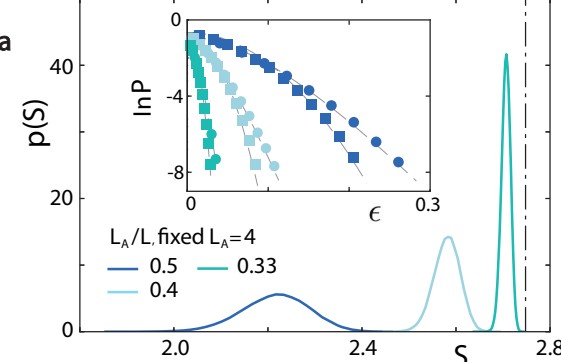

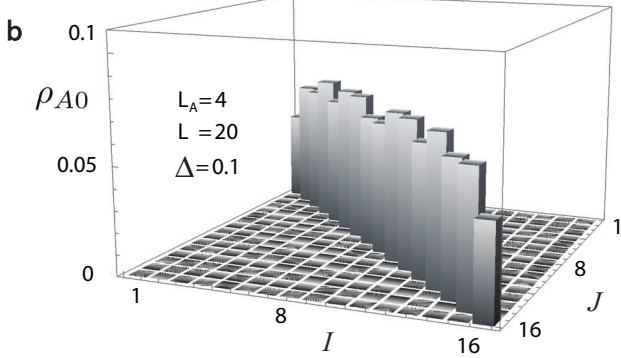

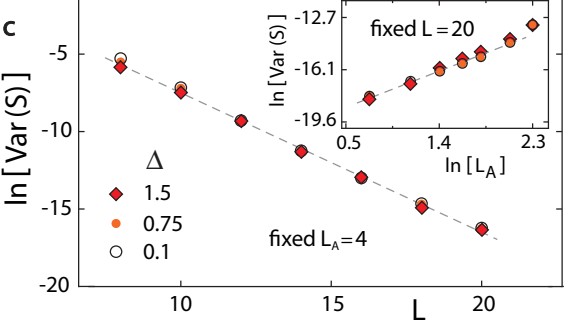

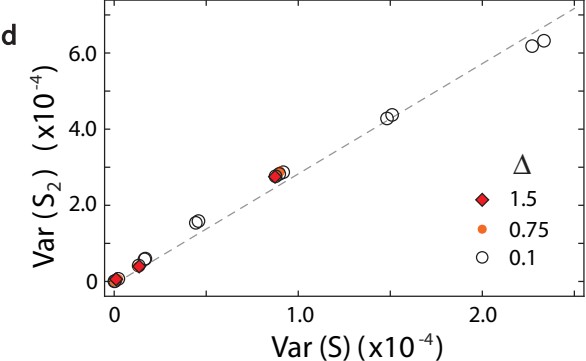

**Fig. 5 | Entanglement fluctuations of _XXZ_ model.** We perform long-time simulations for this model to study the mesoscopic fluctuations in entanglement evolution. Initially, the system is at an unpolarized random state. **a** Main panel: The statistical distribution of entanglement entropy for different ratios of $L_A/L$. Inset: The large deviation probability $\mathbf{P}(|S - \langle S\rangle| \geq \epsilon)$, with upper (squares) and lower (circles) tail respectively, is well fitted by Eq. (2) (dashed lines). $\Delta = 0.1$. **b** Calculating the matrix elements $(\rho_{A0})_{IJ}$ numerically, it is confirmed that $\rho_{A0}$ is proportional to the unit matrix. **c** The simulated Var($S$) (symbols) follows the scaling law described by the second line of Eq. (1) (dashed lines). In the main panel (inset), $L_A$ ($L$) is fixed. **d** For different $\Delta$ and small $L_A/L$, simulations show that Var($S$) $\propto$ Var($S_2$).

account, we find that $\tilde{\rho}_{A1}(\boldsymbol{\varphi}) \sim \sqrt{L_A^\mu / D^{1-\nu}}$. So the disorder strength decays exponentially with $L$.

Uncovering this random structure, we can also use the modified logarithmic Sobolov inequality (7) to address fluctuation statistics. Repeating previous analysis, we find that the distribution of $S$, $S_2$, etc. all follow Eq. (2), as confirmed numerically (Fig. 5a). And similar to noninteracting models, the variances of probes $O = S$, $S_2$, etc. satisfy Eq. (12). Moreover, as shown in Supplementary Note 9 and confirmed numerically (Fig. 5b), $\rho_{A0} \approx \mathbb{I}/D_A$ for $1 \ll L_A \ll L$, with $D_A$ being the dimension of the subsystem Hilbert space: This belongs to the class of concentration-of-measure phenomena[16,45,54,55], investigations of whose relations to quantum entanglement were initiated in ref. [16]. Substituting this $\rho_{A0}$ into Eq. (12) we find that

$$\text{Var}(O) \propto D_A^2 \langle |\text{Tr}_A(\tilde{\rho}_{A1}\partial_{\boldsymbol{\varphi}}\tilde{\rho}_{A1})|^2 \rangle, \tag{17}$$

where the proportionality coefficient is independent of $L$, $L_A$. Thus the scaling behaviors of mesoscopic fluctuations of entanglement dynamics must be independent of the choice of $O$, as confirmed numerically (d). Substituting Eq. (16) into Eq. (17), we find $\text{Var}(O) \sim L_A^\beta$ $e^{-\kappa L}$ ($\kappa > 0$, $\beta = 2\mu \in [2,4]$). After rescaling $L$, $L_A$ and $\text{Var}(O)$, we obtain the second line of Eq. (1). In simulations, $\text{Var}(O)$ is indeed seen to decay exponentially with $L$ for fixed $L_A$ (c, main panel), and to display a power-law increase in $L_A$ for fixed $L$ (c, inset), with a power 2.4.

## Discussion

Our theory essentially hinges upon the relation between the wave-function evolution and the trajectory $\boldsymbol{\varphi} = \boldsymbol{\omega}t$ on a high-dimensional torus, and the information-theoretic observable as a function on that torus. So it can be extended to more general contexts. First, it applies to other characteristics of entanglement especially the subsystem complexity, studies of whose temporal fluctuations have been initiated recently[58]. Second, our ongoing studies have shown that there are no principal difficulties in generalizing the present theory to nonintegrable interacting spin chains. Third, experiments show that the entanglement dynamics of Bose–Hubbard model exhibit temporal fluctuations[6]; it is interesting to generalize the present results to that model and compare them with existing measurements. Fourth, our ongoing studies have also shown that the general theory developed for interacting models, while based on the full reduced density of the matrix, should be capable of unifying results for noninteracting and interacting models. Indeed, for noniteracting models, provided the initial state is Gaussian we can use that theory to reproduce Eq. (1); for the non-Gaussian initial state, we can show that the emergent disorder carries the same structure as in the Gaussian case, which is a key leading to the scaling law (1) and suggests its robustness. Finally, because each virtual disordered sample corresponds to a pure state, our work suggests a simple way of producing a random pure-state ensemble to which great experimental efforts[59] are made. That is, we evolve an initial pure state by a single Hamiltonian and collect states at distinct sufficiently long times.

## Data availability

All data are displayed in the main text and Supplementary Information. Additional data are available from the corresponding author upon request.

## Code availability

The code that supports the plots in the main text and Supplementary Information is available from the corresponding author upon request.

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

## Acknowledgements

We thank Italo Guarneri and Xin Wan for discussions, and Jean-Claude Garreau and Azriel Z. Genack for comments on the manuscript. This work is supported by the National Natural Science Foundation of China Grant Nos. 11925507 (C.T.), 12047503 (C.T.), and 11974308 (L.-K.L.).

## Author contributions

L.-K.L. and C.T. designed the project and wrote the manuscript. The mathematical ideas were conceived by C.T. and developed into a full theory by C.T. and L.-K.L. together. Numerical analysis was conducted by L.-K.L. and C.L.

## Competing interests

The authors declare no competing interests.
