## [Peer Review File · Nature Communications]

REVIEWER COMMENTS

Reviewer #1 (Remarks to the Author):

In this paper, the authors study entanglement fluctuations in a class of free fermionic chains out of equilibrium. The focus is on finite chains, and fluctuations over large times (no disorder). It is nevertheless argued that the results can be reproduced by a simple model with disordered samples. Entanglement fluctuations are then bounded by making use of concentration inequalities, and a numerical comparison with the actual time evolution is made.

Overall this is an interesting paper, and the results appear to be correct. Two limitations are the fact that the system studied are free fermionic (besides being just integrable, as insisted on in the text). Since the study also involves extremely long times, it is also not clear how such effects could be observed in practice, or how integrability breaking effects can be neglected in a realistic system.

For these reasons, the paper is in my opinion not sufficiently groundbreaking to warrant publication in Nature Communications. I would recommend publication in a more specialized journal.

The paper is reasonably clear and well written. However the various quench protocols could be better explained in the main text, since reading the various figures necessitates a little bit too much reading through the supplemental material.

Reviewer #2 (Remarks to the Author):

In this paper, the authors study entanglement dynamics of a class of integrable models including the Rice-Mele model and the XX spin chain (which the authors call the transverse field Ising chain). These models are essentially free models. As a result, 1) the correlation matrix can be computed analytically in terms of Toeplitz type matrices; 2) entanglement entropy can be related to the correlation matrix in a simple way, given by Eq(3) in the main text. In the quench protocol, the initial state is taken to be a Gaussian state. The authors study the time evolution of the entanglement entropy of such states.

Since time dependence of the correlation matrix only enter through the dynamical phases, the correlation matrix can be considered as a function on the N -dimensional torus. This allows one to introduce a statistical interpretation for the long time dynamics of entanglement measures, and exploit techniques from modern probability theory. The key point is the statistical equivalence between out-of-equilibrium and sample-to-sample fluctuations of various entanglement probes. From the statistical interpretation, the authors obtained the following results

(a) Spectral statistics of the correlation matrix are investigated, revealing a distinctive level repulsion phenomenon typically associated with quantum chaotic systems. The authors claim that this phenomena is surprising because the starting system is free and deterministic, yet the entanglement dynamics exhibit chaotic features.

(b) The authors study the statistics of the entanglement entropy and found that the variance exhibits a universal scaling law. The probability for the large deviation is asymmetric with respect to its mean, with a sub-Gaussian upper and sub-Gamma lower tail.

Based on the above findings, alongside the resemblance of the time fluctuations to mesoscopic fluctuations, the authors suggest the utilization of quantum information theoretic measures as unorthodox mesoscopic probes. This insight, linking fluctuations with time and fluctuation with disordered samples, holds potential beyond the immediate context. The outcomes, in my assessment, is both valuable and solid.

However, a notable concern is the following. These findings seem to hinge heavily on the specific characteristic that time dependence enters exclusively through dynamical phases, enabling the correlation matrix's representation as a function on the N -torus. This reliance implies sensitivity to both the model and the initial state. While the current results are certainly notable, their universal applicability might be contingent upon the specificities of these chosen factors. This situation raises the possibility that the outcomes, despite their elegance, might remain isolated and devoid of broader universal significance, potentially constituting an artifact of the inherent simplicity of the free systems studied here.

That being said, I would recommend the paper for publication if the authors could provide convincing evidence that the ensemble interpretation for entanglement dynamics is also valid for more general situations, either by changing the initial states of the current models or even better, by considering interacting integrable systems like the Heisenberg XXX or XXZ spin chain.

Reviewer #3 (Remarks to the Author):

In this manuscript, the authors study the random structure of the entanglement due to the dynamics of the wavefunction after a global quantum quench. Similar studies have been done at equilibrium, considering a random sampling from a pure state ensemble, while here the authors are interested in the random structure that the evolution of the wavefunction imprints into the correlation matrix $C(t)$. To the best of my knowledge, a similar analysis was initiated in arxiv:2102.02764 for the entanglement entropy and the subsystem complexity, but focusing on a time regime different with respect to the current paper, where a quasiparticle picture inside each revival of the entanglement was still valid. Another crucial difference with respect to 'standard' studies about the entanglement dynamics is the fact that the total system size is finite, and this is what leads to N different dynamical phases that endow $C(t)$ with a random structure.

The result of this analysis strictly relies on the relation between the entanglement entropy and $C(t)$, so it naturally applies to Gaussian states. The authors find the analytical behaviour of the probability for large deviation of S from $\langle S \rangle$, as well as a universal scaling of the variance of S as a function of the (sub)system size, which they can explain through the constructive interference between the paths of the entangled quasiparticle pairs.

Despite the results being very interesting, well explained, and supported by analytical and numerical computations also in the supplemental material, the main obstruction to their publication in Nature Communications is that the results seem to be a prerogative of free theories, and the generalisation to non-Gaussian theories is not immediate starting from the methods used in the paper. It would be very interesting if the authors could quantitatively discuss this in their paper, also because studying the late-time regime of non-Gaussian theories is tough even from a numerical point of view (e.g. one could use exact diagonalisation, but then this would give a constraint on the maximal (sub)system size one can reach). Without major overhaul/additional works to justify the generality, I would not recommend publication in Nature Communications.

REPLY TO REVIEWER #1

Reviewer: *In this paper, the authors study entanglement fluctuations in a class of free fermionic chains out of equilibrium. The focus is on finite chains, and fluctuations over large times (no disorder). It is nevertheless argued that the results can be reproduced by a simple model with disordered samples. Entanglement fluctuations are then bounded by making use of concentration inequalities, and a numerical comparison with the actual time evolution is made.*

Overall this is an interesting paper, and the results appear to be correct.

Answer: We thank the Reviewer for refereeing our work and the positive remark that our manuscript is “*an interesting paper*” and “*results appear to be correct*”.

Reviewer: *Two limitations are the fact that the system studied are free fermionic (besides being just integrable, as insisted on in the text).*

Answer: We agree with the Reviewer’s main concern which all three Reviewers have raised in common, namely, what happens when interactions present. We thank the Reviewer for this critical comment. Addressing this issue has led to substantial work and results that prove to be far more general than our original study.

In the revised manuscript (see new section on Results, Class II models, Fig. 5 in the main text, and Sec. IX in Supplemental Information for details), we extended our original study to a paradigmatic model of interaction, namely the Heisenberg XXZ spin chain suggested by Reviewer #2, that is integrable and interacting, by generalizing the analytical theory developed for free fermions and by exact diagonalization numerical technique.

First of all, we uncover the random structure hidden behind entanglement dynamics of interacting spin chains. Specifically, the entanglement dynamics for the interacting problem is encoded in the time-dependent reduced density matrix (RDM). The important point is the overall structure of its time dependence: the time parameter enters as dynamical phase factors as in our original study, but associated with many-body eigenfrequencies. The number of independent incommensurate frequencies is now determined by the effective number of eigenbases excited by the initial state $|\mathcal{D}\rangle$, which increases exponentially with the total system size L .

Then, with $|\mathcal{D}\rangle$ incommensurate frequencies, the time evolution induces a motion of RDM on the $|\mathcal{D}\rangle$ -torus, with the ensuing trajectory being dense. Therefore, by ergodicity, the time series is statistically equivalent to a uniform sampling of an ensemble of RDM on the $|\mathcal{D}\rangle$ -torus. By following essentially the same analytical derivation as the random correlation matrix method (for free fermions), we arrive at extended results for the interacting spin chains summarized in the following:

- (1) mesoscopic fluctuations persist to arbitrarily long time (Fig 5(a));
- (2) asymmetric statistical distribution flanked by sub-Gamma lower tail and sub-Gaussian upper tail (Fig 5(a) inset);
- (3) exponential (with L) and power law (with the subsystem size L_A) scaling behaviours for the fluctuations (Fig 5(c));
- (4) universality with respect to different entanglement probes, i.e., entanglement entropy and Renyi entropy of different orders (Fig 5(d)).

Finally, we verify our results by exact diagonalization numerical technique (with the full many-body eigenstates and spectrum) with various forms of initial state (Neel state, and two forms of random states) for L up to 20.

Reviewer: *Since the study also involves extremely long times, it is also not clear how such effects could be observed in practice,*

Answer: In fact, temporal fluctuations have been observed in previous quantum simulation experiments on entanglement dynamics:

(a) Greiner’s group, Harvard University:

A. M. Kaufman *et al*, Quantum thermalization through entanglement in an isolated many-body system, Science 353, 794 (2016), page 796, third column,

“Initially, we observed an approximately linear rise in the entropy with time, with a similar slope among the subsystems considered (Fig. 3, inset) (2). After an amount of time that depended on the subsystem size, the entanglement entropy saturated to a steady-state value, about which there were small residual temporal fluctuations. The presence of residual fluctuations is attributable in part to the finite size of our system.”

(b) Martinis’s group, UC Santa Barbara:

C. Neill et al, Ergodic dynamics and thermalization in an isolated quantum system, Nature Physics 12, 1037 (2016), page 1038, second column,

“In Fig. 2a,b, we see that the entropy fluctuates over time, In small quantum systems, there are fluctuations or revivals that vanish when the system size is taken to infinity (known as the thermodynamic limit). For finite systems, averaging the entropy over time is commonly used to estimate the equilibrium value approached by larger systems.”

With our original study now generalized to interacting models, our work has intimate relations with these observations. Indeed, mesoscopic entanglement fluctuations in quench dynamics have been observed in (a) cold atoms and (b) superconducting qubits quantum simulators. For cold atoms simulating the Bose-Hubbard model, the observation time extending to tens of milliseconds shows the importance of entanglement fluctuations with increasing subsystem size for a fixed total system (see Fig. 3 of experiment (a)). Similarly, in the superconducting qubits experiment (b), entanglement fluctuations over time are also observed.

To further clarify the issue, we have added results of short time simulation of the Heisenberg XXZ model with a random initial state in the Supplemental Information, see Fig S6. We demonstrate a rapid approach to the mesoscopic fluctuating regime in entanglement evolution in interacting spin models. Consequently, scaling behaviour is found even for a short evolution time.

Reviewer: *or how integrability breaking effects can be neglected in a realistic system.*

Answer: We have verified effects of nonintegrability in spin chains, either with an external magnetic field or with next-nearest-neighbor couplings by exact diagonalization numerical simulation. They do not change our conclusions for the integrable interacting spin models in any qualitative way.

Reviewer: *For these reasons, the paper is in my opinion not sufficiently groundbreaking to warrant publication in Nature Communications. I would recommend publication in a more specialized journal.*

Answer: With substantial revisions, we have addressed the generality of mesoscopic entanglement fluctuations phenomenon in both free and interacting systems. Moreover, we presented clear evidence of mesoscopic entanglement fluctuations in current experiments, opening the avenue to probe the characteristic distribution. We hope the Reviewer could agree now that our revised manuscript has wider implications than the original study, and has answered an important question regarding quantum dynamics of information-theoretic observables as well as quantum thermalization, therefore suitable for publication in Nature Communications.

Reviewer: *The paper is reasonably clear and well written. However the various quench protocols could be better explained in the main text, since reading the various figures necessitates a little bit too much reading through the supplemental material.*

Answer: We thank the Reviewer for the positive remark and suggestion. We added sentences in explaining the quench protocols in the captions in Figures 1-3 in the main text.

REPLY TO REVIEWER #2

Reviewer: *In this paper, the authors study entanglement dynamics of a class of integrable models including the Rice-Mele model and the XX spin chain (which the authors call the transverse field Ising chain). These models are essentially free models. As a result, 1) the correlation matrix can be computed analytically in terms of Toeplitz type matrices; 2) entanglement entropy can be related to the correlation matrix in a simple way, given by Eq(3) in the main text. In the quench protocol, the initial state is taken to be a Gaussian state. The authors study the time evolution of the entanglement entropy of such states.*

Since time dependence of the correlation matrix only enter through the dynamical phases, the correlation matrix can be considered as a function on the S^1 -dimensional torus. This allows one to introduce a statistical interpretation for the long time dynamics of entanglement measures, and exploit techniques from modern probability theory. The key point is the statistical equivalence between out-of-equilibrium and sample-to-sample fluctuations of various entanglement probes. From the statistical interpretation, the authors obtained the following results

(a) Spectral statistics of the correlation matrix are investigated, revealing a distinctive level repulsion phenomenon typically associated with quantum chaotic systems. The authors claim that this phenomena is surprising because the starting system is free and deterministic, yet the entanglement dynamics exhibit chaotic features.

(b) The authors study the statistics of the entanglement entropy and found that the variance exhibits a universal scaling

law. The probability for the large deviation is asymmetric with respect to its mean, with a sub-Gaussian upper and sub-Gamma lower tail.

Based on the above findings, alongside the resemblance of the time fluctuations to mesoscopic fluctuations, the authors suggest the utilization of quantum information theoretic measures as unorthodox mesoscopic probes. This insight, linking fluctuations with time and fluctuation with disordered samples, holds potential beyond the immediate context. The outcomes, in my assessment, is both valuable and solid.

Answer: We thank the Reviewer for refereeing our work. We highly appreciate his/her detailed assessment covering the full scope of our work. We further thank the Reviewer for finding our work “both valuable and solid”. As we explain in details below, the substantially revised manuscript further lays down a solid basis for the Reviewer’s remark on our treatment of free fermions which “holds potential beyond the immediate context”.

Reviewer: However, a notable concern is the following. These findings seem to hinge heavily on the specific characteristic that time dependence enters exclusively through dynamical phases, enabling the correlation matrix’s representation as a function on the \mathbb{S}^1 -torus. This reliance implies sensitivity to both the model and the initial state. While the current results are certainly notable, their universal applicability might be contingent upon the specificities of these chosen factors. This situation raises the possibility that the outcomes, despite their elegance, might remain isolated and devoid of broader universal significance, potentially constituting an artifact of the inherent simplicity of the free systems studied here.

That being said, I would recommend the paper for publication if the authors could provide convincing evidence that the ensemble interpretation for entanglement dynamics is also valid for more general situations, either by changing the initial states of the current models or even better, by considering interacting integrable systems like the Heisenberg XXX or XXZ spin chain.

Answer: We agree with the Reviewer’s main concern which all three Reviewers have raised in common, namely, what happens when interactions are present. Addressing this key issue has significantly levelled up our original study.

In the revised manuscript (see new section on Results, Class II models, Fig. 5 in the main text, and Sec. IX in Supplemental Information for details), following the Reviewer’s suggestion, we extended our original study to a paradigmatic model of interaction, namely the Heisenberg XXZ spin chain that is integrable and interacting, by generalizing the analytical theory developed for free fermions and by exact diagonalization numerical technique.

First of all, we uncover the random structure hidden behind entanglement dynamics of interacting spin chains. Specifically, the entanglement dynamics for the interacting problem is encoded in the time-dependent reduced density matrix (RDM). The important point is the overall structure of its time dependence: the time parameter enters as dynamical phase factors as in our original study, but associated with many-body eigenfrequencies. The number of independent incommensurate frequencies is now determined by the effective number of eigenbases excited by the initial state ρ , which increases exponentially with the total system size L .

Then, with ρ incommensurate frequencies, the time evolution induces a motion of RDM on the \mathbb{S}^1 -torus, with the ensuing trajectory being dense. Therefore, by ergodicity, the time series is statistically equivalent to a uniform sampling of an ensemble of RDM on the \mathbb{S}^1 -torus. By following essentially the same analytical derivation as the random correlation matrix method (for free fermions), we arrive at extended results for the interacting spin chains summarized in the following:

- (1) mesoscopic fluctuations persist to arbitrarily long time (Fig 5(a));
- (2) asymmetric statistical distribution flanked by sub-Gamma lower tail and sub-Gaussian upper tail (Fig 5(a) inset);
- (3) exponential (with L) and power law (with the subsystem size L_A) scaling behaviours for the fluctuations (Fig 5(c));
- (4) universality with respect to different entanglement probes, i.e., entanglement entropy and Renyi entropy of different orders (Fig 5(d)).

The results presented in Fig. 5 are obtained by exact diagonalization numerical technique (with the full many-body eigenstates and spectrum) with a random initial state for L up to 20. In the Supplemental Information, extended numerical results are presented, including the simulation with a Neel initial state (Fig. S5), as well as a demonstration of convergence to scaling behaviours even for short time entanglement evolution with a random initial state (Fig. S6).

With substantial revisions, we have addressed the generality of mesoscopic entanglement fluctuations phenomenon in both free and interacting systems. We hope the Reviewer could agree now that our revised manuscript has wider implications than the original study, and has answered an important question regarding quantum dynamics of

information-theoretic observables as well as quantum thermalization, therefore suitable for publication in Nature Communications.

REPLY TO REVIEWER #3

Reviewer: *In this manuscript, the authors study the random structure of the entanglement due to the dynamics of the wavefunction after a global quantum quench. Similar studies have been done at equilibrium, considering a random sampling from a pure state ensemble, while here the authors are interested in the random structure that the evolution of the wavefunction imprints into the correlation matrix $\mathcal{C}(t)$. To the best of my knowledge, a similar analysis was initiated in arxiv:2102.02764 for the entanglement entropy and the subsystem complexity, but focusing on a time regime different with respect to the current paper, where a quasiparticle picture inside each revival of the entanglement was still valid. Another crucial difference with respect to 'standard' studies about the entanglement dynamics is the fact that the total system size is finite, and this is what leads to N different dynamical phases that endow $\mathcal{C}(t)$ with a random structure.*

Answer: We thank the Reviewer for refereeing our work. We also thank the Reviewer for informing us on the complexity study in quench dynamics in bosonic chains, which we have added as Ref. [57] in the revised manuscript.

Reviewer: *The result of this analysis strictly relies on the relation between the entanglement entropy and $\mathcal{C}(t)$, so it naturally applies to Gaussian states. The authors find the analytical behaviour of the probability for large deviation of \mathcal{C} from $\langle \mathcal{C} \rangle$, as well as a universal scaling of the variance of \mathcal{C} as a function of the (sub)system size, which they can explain through the constructive interference between the paths of the entangled quasiparticle pairs.*

Despite the results being very interesting, well explained, and supported by analytical and numerical computations also in the supplemental material, the main obstruction to their publication in Nature Communications is that the results seem to be a prerogative of free theories, and the generalisation to non-Gaussian theories is not immediate starting from the methods used in the paper. It would be very interesting if the authors could quantitatively discuss this in their paper, also because studying the late-time regime of non-Gaussian theories is tough even from a numerical point of view (e.g. one could use exact diagonalisation, but then this would give a constraint on the maximal (sub)system size one can reach). Without major overhaul/additional works to justify the generality, I would not recommend publication in Nature Communications.

Answer: We thank the Reviewer for the positive remarks on our previous manuscript. As with all three reviewers, a crucial point raised is what happens when interactions are present. Addressing this key issue has significantly levelled up our original study.

In the revised manuscript (see new section on Results, Class II models, Fig. 5 in the main text, and Sec. IX in Supplemental Information for details), we extended our original study to a paradigmatic model of interaction, namely the Heisenberg XXZ spin chain as suggested by Reviewer #2, that is integrable and interacting, by generalizing the analytical theory developed for free fermions and by exact diagonalization numerical technique.

First of all, we uncover the random structure hidden behind entanglement dynamics of interacting spin chains. Specifically, the entanglement dynamics for the interacting problem is encoded in the time-dependent reduced density matrix (RDM), which is much more general than the correlation matrix in our original study. The important point is the overall structure of its time dependence: the time parameter enters as dynamical phase factors of RDM, and is associated with many-body eigenfrequencies. The number of independent incommensurate frequencies is now determined by the effective number of eigenbases excited by the initial state ρ , which increases exponentially with the total system size L .

Then, with ρ incommensurate frequencies, the time evolution induces a motion of RDM on the ρ -torus, with the ensuing trajectory being dense. Therefore, by ergodicity, the time series is statistically equivalent to a uniform sampling of an ensemble of RDM on the ρ -torus. By following essentially the same analytical derivation as the random correlation matrix method (for free fermions), we arrive at extended results for the interacting spin chains summarized in the following:

- (1) mesoscopic fluctuations persist to arbitrarily long time (Fig 5(a));
- (2) asymmetric statistical distribution flanked by sub-Gamma lower tail and sub-Gaussian upper tail (Fig 5(a) inset);
- (3) exponential (with L) and power law (with the subsystem size L_A) scaling behaviours for the fluctuations (Fig 5(c));
- (4) universality with respect to different entanglement probes, i.e., entanglement entropy and Renyi entropy of different orders (Fig 5(d)).

The results presented in Fig. 5 are obtained by exact diagonalization numerical technique (with the full many-body eigenstates and spectrum) with random initial state for L up to 20. In the Supplemental Information, extended numerical results are presented, including the simulation with a Neel initial state (Fig. S5), as well as a demonstration of convergence to scaling behaviours even for short time entanglement evolution with a random initial state (Fig. S6).

With substantial revisions, we have addressed the generality of mesoscopic entanglement fluctuations phenomenon in both free and interacting (i.e., non-Gaussian) systems. We hope the Reviewer could agree now that our revised manuscript has wider implications than the original study, and has answered an important question regarding quantum dynamics of information-theoretic observables as well as quantum thermalization, therefore suitable for publication in Nature Communications.

REVIEWERS' COMMENTS

Reviewer #1 (Remarks to the Author):

I acknowledge the authors made a significant effort to address my concerns, as well as the other referees'. I also do agree that the findings presented on the XXZ spin chain are interesting.

I believe the presentation should be improved, as some of the discussions are confusing. Provided this is fixed and the final result are still convincing, the paper might be publishable in Nature Communication.

1) First the use of the notion of integrability. The arguments used to study the XXZ model do not make any use of integrability, so probably have nothing to do with it. The authors themselves acknowledge this in the conclusion ('our on-going studies have shown that there are no principal difficulties to generalize the present theory to nonintegrable interacting spin chain'), so the important distinction is between free systems and interacting systems. For this reason I find the terminology with class I and class II models to be a little bit strange.

2) Since the XXZ model also has a free point ($\Delta=0$), it might be useful to show this data somewhere in figure 5, to contrast with previous results in the paper. As a more minor comment, the data for $\Delta=0.1$ is hidden by the data points for $\Delta=0.75$ especially in figure 5 (c).

3) Nothing prevents me from studying also the full reduced density matrix for free models, invoking also the IPR to estimate D. What happens in that case, and also how much of it depends on the initial state? Such a discussion would help related the two types of models studied in this paper.

Reviewer #2 (Remarks to the Author):

In the revised version, the authors have made significant improvement of the paper. Previously, the authors' observation seem to depend heavily on the fact that they are working on essentially free

systems. In the current version, they extended the study to the XXZ spin chain which is interacting and integrable. They also find random structure hidden behind the entanglement dynamics of this integrable spin chain. This gives further evidence that the random structure in entanglement dynamics is a general phenomena. With these modifications, I'm glad to recommend the paper to be published.

Reviewer #3 (Remarks to the Author):

The major revision in this submission is the extension to the interacting XXZ model. The additional numerical evidences in Fig. 5 show that the variance of the entanglement exponentially decays with system size (rather than polynomially, as it happens in the free case). The universal distribution of the entanglement fluctuates also at long times and these fluctuations are described by Eq. 2, as in the free case. The thorough revision have cleared my concerns and I recommend the manuscript for acceptance.

I only have one more technical question: in Fig. 5, the authors have considered a quench to values of Δ between -1 and 1, i.e. in the critical regime of the XXZ spin chain. Have they noticed any differences for values of Δ outside this region?

REPLY TO REVIEWER #1

Reviewer: *I acknowledge the authors made a significant effort to address my concerns, as well as the other referees'. I also do agree that the findings presented on the XXZ spin chain are interesting.*

I believe the presentation should be improved, as some of the discussions are confusing. Provided this is fixed and the final result are still convincing, the paper might be publishable in Nature Communication.

Answer: We thank the Reviewer for refereeing our work again. In particular, we would like to thank him/her for the comments below, which help us to further improve the presentation. We are glad that he/she agrees that “Provided this is fixed and the final result are still convincing, the paper might be publishable in Nature Communication.”

Reviewer: *1) First the use of the notion of integrability. The arguments used to study the XXZ model do not make any use of integrability, so probably have nothing to do with it. The authors themselves acknowledge this in the conclusion ('our ongoing studies have shown that there are no principal difficulties to generalize the present theory to nonintegrable interacting spin chain'), so the important distinction is between free systems and interacting systems. For this reason I find the terminology with class I and class II models to be a little bit strange.*

Answer: We have changed the terminology “class I model” to “noninteracting model” and “class II model” to “interacting model”.

Reviewer: *2) Since the XXZ model also has a free point ($\Delta=0$), it might be useful to show this data somewhere in figure 5, to contrast with previous results in the paper.*

Answer: We fully agree it would be useful to compare the free point result in Fig. 5. As we already tried before, we find the presentation non-ideal. The reasons are as follows. The power law scaling behavior for the free point $\Delta=0$ appears only for large sizes L, L_A (cf, see Fig. 3). So on the length scale used for Fig. 5, it only appears as a gradual crossover, as $\Delta \rightarrow 0$, from the exponential behavior shown (for finite Δ) to a power law behavior on the semi-log scale. On the other hand, obtaining data for nonvanishing Δ values for such large sizes as in Fig. 3 is beyond numerical reach.

Reviewer: *As a more minor comment, the data for $\Delta=0.1$ is hidden by the data points for $\Delta=0.75$ especially in figure 5 (c).*

Answer: In the revised manuscript, we have made a minor correction to Fig. 5 (c,d) to make the data points for various Δ values distinguishable.

3) Nothing prevents me from studying also the full reduced density matrix for free models, invoking also the IPR to estimate D . What happens in that case, and also how much of it depends on the initial state? Such a discussion would help related the two types of models studied in this paper.

Answer: We thank the Reviewer #1 for this question. For the initial state being Gaussian, we can reproduce the result namely Eq. (1) for free models by using the full reduced density of matrix, or more precisely the general method described in the second part of our manuscript. For the initial state being non-Gaussian, it is much more difficult, both analytically and numerically. Nevertheless, we have been able to show that the emergent disorder structure is exactly the same as the Gaussian case, which is a key leading to Eq. (1). Our preliminary analytical calculations further show that (at least) for some non-Gaussian initial states Eq. (1) does hold. These analysis indicate that for free models the scaling behaviors are insensitive to the initial state.

In the revised manuscript we added a discussion on this in the Discussion section.

REPLY TO REVIEWER #3

Reviewer: *The major revision in this submission is the extension to the interacting XXZ model. The additional numerical evidences in Fig. 5 show that the variance of the entanglement exponentially decays with system size (rather than polynomially, as it happens in the free case). The universal distribution of the entanglement fluctuates also at long times and these fluctuations are described by Eq. 2, as in the free case. The thorough revision have cleared my concerns and I recommend the manuscript for acceptance.*

I only have one more technical question: in Fig. 5, the authors have considered a quench to values of Δ between -1 and 1, i.e. in the critical regime of the XXZ spin chain. Have they noticed any differences for values of Δ outside this region?

Answer: No, we haven't noticed any differences for values of Δ outside that region. In Fig. 5 of the revised manuscript, we added the results for $\Delta=1.5$ to show this. For negative Δ data falling on the same line, we do not show them again.